# Translating Virtual Reality Cue Exposure Therapy for Binge Eating into a Real-World Setting: An Uncontrolled Pilot Study

**DOI:** 10.3390/jcm10071511

**Published:** 2021-04-05

**Authors:** Katherine Nameth, Theresa Brown, Kim Bullock, Sarah Adler, Giuseppe Riva, Debra Safer, Cristin Runfola

**Affiliations:** 1Department of Psychiatry and Behavioral Sciences, 401 Quarry Rd, Stanford University, Stanford, CA 94305, USA; kbullock@stanford.edu (K.B.); sadler1@stanford.edu (S.A.); dlsafer@stanford.edu (D.S.); 2PGSP-Stanford PsyD Consortium, 1791 Arastradero Rd, Palo Alto, CA 94304, USA; tbrown@paloaltou.edu; 3Applied Technology for Neuro-Psychology Lab. Istituto Auxologico Italiano, 20095 Milan, Italy; giuseppe.riva@unicatt.it; 4Centro Studi e Ricerche di Psicologia della Comunicazione, Università Cattolica del Sacro Cuore, 20123 Milan, Italy

**Keywords:** eating disorder, binge-eating disorder, bulimia nervosa, binge eating, cue-exposure, therapy, treatment, virtual reality

## Abstract

Binge-eating disorder (BED) and bulimia nervosa (BN) have adverse psychological and medical consequences. Innovative interventions, like the integration of virtual reality (VR) with cue-exposure therapy (VR-CET), enhance outcomes for refractory patients compared to cognitive behavior therapy (CBT). Little is known about the feasibility and acceptability of translating VR-CET into real-world settings. To investigate this question, adults previously treated for BED or BN with at least one objective or subjective binge episode/week were recruited from an outpatient university eating disorder clinic to receive up to eight weekly one-hour VR-CET sessions. Eleven of 16 (68.8%) eligible patients were enrolled; nine (82%) completed treatment; and 82% (9/11) provided follow-up data 7.1 (SD = 2.12) months post-treatment. Overall, participant and therapist acceptability of VR-CET was high. Intent-to-treat objective binge episodes (OBEs) decreased significantly from 3.3 to 0.9/week (*p* < 0.001). Post-treatment OBE 7-day abstinence rate for completers was 56%, with 22% abstinent for 28 days at follow-up. Among participants purging at baseline, episodes decreased from a mean of one to zero/week, with 100% abstinence maintained at follow-up. The adoption of VR-CET into real-world clinic settings appears feasible and acceptable, with a preliminary signal of effectiveness. Findings, including some loss of treatment gains during follow-up may inform future treatment development.

## 1. Introduction

Binge-eating disorder (BED) and bulimia nervosa (BN) are eating disorders characterized by binge eating, including recurrent episodes of eating objectively large amounts of food accompanied by a loss of control (LOC) [1]. Both disorders are associated with severe adverse psychological and medical consequences [2,3], an increased risk of death [4,5], and high public health costs [6]. BED is the most common eating disorder in the U.S. [7].

Cognitive behavioral therapy (CBT) is considered the gold standard treatment for BED and BN [8,9,10]. Recent meta-analyses show that despite CBT’s efficacy, up to 50–60% of patients with BED and BN do not fully respond to treatment [11,12]. In addition, long-term recovery is often not sustained [13]. Alternative options are needed to improve outcomes and sustain recovery, especially for those who fail initial treatment.

Virtual reality (VR) is a technological tool that can supplement evidence-based treatments with the possibility of enhancing outcomes. VR allows patients to naturally interact with computer-generated experiences and stimuli representing the real world while simultaneously benefiting from a clinical, supervised setting [14]. There are a few different types of immersion displays. Immersive VR uses head mounted displays (HMDs), allowing patients to be completely immersed in the VR environment and isolated from the real world [14,15]. Non-immersive or semi-immersive VR may use 3D laptops and 3D vision via polarized glasses [14,16]. With technological advancements, new generations of VR systems have significantly reduced costs, making immersive VR more accessible. Currently, most literature reviews on VR-based therapies for mental disorders have focused on studies using immersive VR as opposed to non- or semi-immersive VR [14,15,17,18].

European studies demonstrate that the integration of VR (both immersive and semi-immersive) with cue-exposure therapy (VR-CET) for binge-type eating disorders enhances treatment outcomes for refractory patients compared to CBT alone [19,20,21]. VR-CET provides traditional CET in a virtual environment. CET for eating disorders is based on the classical conditioning model of binge eating [22]. According to this model, the intake of food and its metabolic effects are conceptualized as the unconditioned stimulus and the unconditioned response, respectively. Cues that signal food intake (e.g., sight, smell, taste, and the context or environment) act as conditioned stimuli. The presence of these cues (conditioned stimuli) elicits eating-related anxiety and food cravings, which can increase the probability of binge episodes (conditioned response) [23,24,25,26]. CET aims to extinguish the bond between the cues (conditioned stimuli) and the maladaptive binge response (conditioned response) through systematic, gradual exposure to binge cues while binge eating is prevented. Based on the inhibitory learning model, patients create new associations between the conditioned stimuli and the conditioned response that inhibit the existing maladaptive association [27]. For example, as patients are exposed to binge cues and tolerate distress without binging, new associations form—specifically, that the food cues no longer predict a binge. As such, patients may develop new ways of thinking about the cues that are associated with safety, a sense of control, and self-efficacy, as opposed to anxiety or other negative emotions. In VR-CET specifically, patients are repeatedly exposed to binge cues (e.g., virtual foods and food-related environments, such as a cafeteria) to induce cravings and urges to eat while being prevented from binge eating [20]. Research demonstrates that VR food-related environments produce similar emotional, behavioral, and physiological responses as real-life situations in eating disorder patients [28].

VR-CET offers numerous potential advantages to in vivo exposure. First, VR exposures may enhance ecological validity as VR exposures can be tailored to a patient’s fear hierarchy. For example, VR environments may more closely approximate the settings in which problematic eating behaviors take place compared with the clinician’s office or imaginal exercises. Further, a therapist can manipulate a larger number of stimuli (e.g., types of food) within the VR environment than in the real world. The level of customization can help patients to cope with binge cues or complex environments in a safe and controlled setting while maximizing opportunities for inhibitory learning [27,28,29]. Thus, patients may progress faster due to a perception of increased safety and control [28]. In addition, VR-CET may be an effective intermediate step between imaginal exposures and in vivo exposures, which may reduce treatment dropout [28]. The appeal of a new and exciting technology may also contribute to patients’ willingness to engage with exposures in VR. Indeed, a review of VR in the treatment of EDs indicated increased motivation for change in VR treatments and lower rates of loss to follow-up compared with in vivo active comparisons across several studies [28].

Despite the many potential benefits and efficacy research of VR-CET for binge-type eating disorders, there has been minimal adoption of this treatment in the U.S., and it has not been tested in real-world settings to our knowledge [29]. As such, the purpose of this small, uncontrolled pilot study is to examine feasibility, acceptability, and preliminary signals of effectiveness of immersive VR-CET for binge eating in a U.S. real-world eating disorders clinic. This pilot study was informed by Weisz et al.’s 2004 [30] deployment-focused model of intervention development and testing. Weisz and colleagues [30] acknowledge that the gap between efficacy trials and clinical practice (e.g., differences in characteristics of the treated individuals, reasons for seeking treatment, the settings in which treatment is provided, the therapists who provide treatment, the incentive system, etc.) can be so large that interventions unable to accommodate real-world factors may not be successful in practice, no matter the treatment’s robustness under highly controlled research conditions. In order to create the most robust, practice-ready treatments, the deployment-focused model integrates testing of treatments in practice settings early and throughout the treatment development process, rather than as a final phase, to ensure they are applicable to and successful in the settings in which they will be delivered. The initial step of this model is the “development, refinement, pilot testing, and manualizing of the treatment protocol [30]”. Aligned with this initial phase, this uncontrolled pilot study aims to translate a European research-based VR-CET protocol to a culturally adaptive, clinic-ready intervention. We examine the acceptability, feasibility, and preliminary signals of effectiveness of this intervention. Given the pilot nature of this study, there were no a priori hypotheses.

## 2. Materials and Methods

### 2.1. Participant Population

Participants (*n* = 11) were recruited from an outpatient university eating disorders clinic between 3/2019 and 10/2019 via internal referrals by clinic providers who had specialization in eating disorders. All participants had received the clinic’s standard diagnostic evaluation and assessment. The standard diagnostic evaluation was performed by an eating disorder specialist and included an initial assessment to establish an eating disorder diagnosis (when appropriate), any comorbid diagnoses, and the patient’s suitability for outpatient treatment. Study inclusion criteria included: (i) adult women and men at least 18 years of age, (ii) fluency in English, (iii) a previous clinical DSM-5 [1] diagnosis of BED, BN, or other specified feeding or eating disorder (OSFED; e.g., subthreshold BED or BN), (iv) previous eating disorder treatment (defined as any prior therapy experiences focused on targeting eating disorder symptoms), and (v) at least one objective binge episode (OBE) or subjective binge episode (SBE) per week over the past month. Participants were excluded for: (i) alcohol or drug dependence in the past year, (ii) significant suicidal ideation, (iii) severe depression or a developmental disability interfering with functional capacity, (iv) history of psychosis or bipolar I disorder, unless stable on maintenance therapy for at least one year, or (v) a seizure in the past six months. No participants were receiving concurrent psychotherapy for their binge eating or eating disorder during their participation in the intervention. All participants provided informed consent for inclusion before they participated in the study. The study was conducted in accordance with the Declaration of Helsinki, and the protocol was approved by the Institutional Review Board of Stanford University School of Medicine (IRB protocol # = 44849, date of approval = 3/14/2018).

### 2.2. Therapist Population

Study therapists were recruited from an outpatient university eating disorder clinic. An email inviting providers to attend an information session about the use of VR for eating disorders was sent to members of a departmental list-serve focused on eating disorders. The unknown number of active list-serv members included both current and past faculty and trainees. Those who attended the initial meeting were invited to obtain training in the study treatment and use of VR equipment, which consisted of two 2-h workshops. Those who chose to become study therapists were offered ongoing consultation through the clinic’s already existing weekly clinical peer supervision team meeting, or an optional weekly research meeting. Therapists billed clients per standard procedures in the clinic and therefore received RVU credit for providing treatment. Using research funds, therapists were compensated $50/hour for the amount of time spent outside the billed therapy hour (i.e., to complete training and study surveys and set up VR equipment; total $400/participant).

### 2.3. VR-CET Intervention

The VR-CET protocol was originally developed and tested by investigators in Spain and Italy [16,31,32,33]. One of the authors (GS) and a consultant (MFG) answered questions and provided consultation as needed. To ensure this protocol was appropriate for individuals in the U.S., we collected anecdotal data from U.S. eating disorder specialists, and made cultural adaptations to the clinical protocol and VR environments as indicated. We conducted an updated literature search on CET and made adaptations to the protocol based on advances in the science of memory and learning, i.e., conceptualizing exposure effects based on the inhibitory learning model [27,34]. Resulting from this work was the first initial comprehensive VR-CET for binge eating therapist manual, used by study therapists. We will briefly describe the original investigators’ treatment protocol along with our cultural adaptations below.

#### 2.3.1. Structure of Treatment

Treatment consisted of up to eight one-hour VR-CET sessions, consisting of an assessment phase (up to two sessions) followed by an intervention phase (up to six sessions). The original European research protocol delivered the sessions 2/week over 3–4 weeks. In our real-world study, we adapted this schedule to allow participants and therapists the option of either sessions 2/week or sessions 1/week, given this latter frequency of weekly sessions is often more standard for outpatient clinics. All providers and participants chose the 1/week format; as such, the entire intervention was delivered once per week, over 7–8 weeks.

#### 2.3.2. Assessment and Intervention

VR-CET for binge eating uses environments that simulate real-life triggering eating-related situations to help participants change their response to food-related cues to prevent binge eating. Environments include food cues (conditioned stimuli) that provoke a psychophysiological response (e.g., craving and anxiety) known to trigger binge-related eating (conditioned response).

To build the VR-CET program, the original investigators collected data on the most common binge-eating cues (e.g., foods and environments) reported by patients with eating disorders [16,32]. Using virtual reality, they assessed the validity of these foods and environments in terms of eliciting craving and anxiety responses [31]. For the present study, food choices included in the original protocol were adapted for food preferences common to a U.S diet (e.g., baked fish was changed to a hamburger; see Appendix A for our final food list). In addition, the environments, such as the bedroom and diner, were tailored to U.S. norms. However, the total number of foods (30) and environments (4) included in the program were unchanged. These environments were used throughout the therapist-assisted VR-CET sessions. The software used was a Unity-powered manual build of our VR-CET program, created by two study-hired engineering and development companies using research funds. The program was run through VR-compatible MSI laptops and was launched through Steam VR. To view the VR program, participants used the Oculus Rift (HMD), Oculus sensors, and Oculus controllers (see Figure 1).

In the assessment phase, participants were first asked to rank their experience of cravings on a visual analog scale (VAS) from 0 to 100 [35] for 30 different common binge foods and 4 environments (e.g., kitchen, dining room, bedroom, and restaurant/diner; total of 34 scenes). Please refer to Figure 2 for examples; to review all of the virtual foods and environments, refer to the Appendix A). The VAS was presented to participants within the virtual environment so there was no disruption to their immersive experience, and the participants used the Oculus controllers to provide their ratings. This information was used to create a list of the 40 food–environment combinations with the highest cravings, which participants then rated with an anxiety score (i.e., anxiety of losing control over eating) on a VAS from 0 to 100. From this assessment, the VR computer software program created an individualized 13-step exposure hierarchy of the participant’s most anxiety provoking foods/environments to be used in the subsequent intervention sessions.

The up to six intervention sessions utilized the individualized 13-step food–environment hierarchy to begin cue-exposure therapy, starting with the least anxiety provoking food/environment combinations. During each exposure, anxiety was rated by participants using the VAS presented within the virtual environment every 45 s, with participants generally only proceeding to the next exposure upon a 40% reduction in ratings of anxiety about losing control over their eating/urges to binge (as calculated by the computer program). Unlike the original protocol, the therapist had the option to move to the next exposure without a 40% anxiety reduction as clinically indicated. This decision was based on more recent theoretical work regarding mechanisms of exposure therapy, including tolerating versus reducing distressing feelings and urges [27].

Each exposure session included agenda setting, a brief check-in and assessment of binge and/or purge frequency since the last session, homework review, the VR food/environment exposure (30 min at maximum), post-immersion processing, teaching coping skills (optional), and assignment of homework. During the VR exposure, therapists asked participants to immerse themselves as fully as possible by holding and manipulating the food, bringing it to their face, engaging other sensory modalities (e.g., imagining how it would taste), etc. If the anxiety ratings remained low, therapists would check to ensure participants were not under-engaged or demonstrating avoidance behavior. They would do so by asking patients to report on what they were seeing and experiencing while in the virtual environments. If needed, therapists could also increase engagement by using emotional priming. For example, if participants stated that they usually only turned to foods when distressed, therapists collaborated with participants to enhance emotional engagement by eliciting details from recent experiences resulting in distress. In the post-immersion processing, therapists reviewed the VR exposure, discussing the participant’s thoughts, feelings, and reactions to reducing (or tolerating) their anxiety despite not eating to facilitate and reinforce new learning and maximize expectancy violations [34]. Novel to this VR-CET study protocol, but consistent with the CET protocol for overweight adults with binge eating by Boutelle et al. 2017 [36], therapists also had the option of teaching a coping skill to practice out of session. Examples of coping skills [37,38] included (a) changing the physical state of the body (e.g., diaphragmatic breathing and self-soothing via senses); (b) increasing behavioral alternatives to eating (e.g., behavioral activation, mindful urge surfing, and problem solving); (c) changing the attentional focus (e.g., wise mind, distraction, imagery, and self-motivational statements); and (d) enhancing motivation to resist cues (e.g., decision balance and cost–benefit analyses). As such, this intervention included elements of CBT. Homework always involved additional out-of-session exposures, including the request to refrain, for the rest of the day, from eating the food(s) introduced within the VR-CET session to further strengthen disassociation between food triggers and actual behavior. Since inhibitory learning is optimized when exposure is conducted using an assortment of stimuli (in terms of number and type) and methods of approach across as many contexts as possible, therapists used diverse homework exposure assignments to generalize the newly learned behavior [34].

### 2.4. Measures

The following measures were collected via Qualtrics, an online Health Insurance Portability and Accountability Act (HIPAA) compliant survey platform, to determine acceptability, feasibility, and preliminary signal of effectiveness. These data were collected at baseline, post-sessions, post-treatment, and follow-up (e.g., at least 1-month post-treatment).

#### 2.4.1. Descriptive Variables

Age, body mass index (BMI), gender, race, education, eating disorder diagnosis, psychiatric comorbidity, and utilization of concurrent treatment were obtained at baseline.

#### 2.4.2. Feasibility

Outcomes assessing feasibility included: (1) the percentage of providers who, after attending an information session about VR and eating disorders, attended the two 2-h study training workshops; (2) percentage of providers who, after attending the training workshops, subsequently enrolled as study therapists; (3) percentage of eligible (clinician-referred) patients who enrolled in the study; (4) percentage of sessions attended and percentage of study completers; and (5) percentage of questionnaires completed by both participants and study therapists. Definitions of feasibility were based on the research literature [39].

#### 2.4.3. Acceptability

Outcomes assessing acceptability were administered to both participants and therapists. Among participants, acceptability measures included: (1) the simulator sickness questionnaire (SSQ; [40]), a 16-item measure focusing on symptoms of cybersickness (e.g., “Did you experience any psychological symptoms (feeling detached from reality, anxiety, sadness, or any other odd sensations)?”). Its 3 subscales include ratings of: nausea, oculomotor, and disorientation. Possible total scores range from 0 to 235.62, with higher scores indicating a greater degree of cybersickness; (2) presence questionnaire (PQ-revised by 5 items that were not applicable to the study; [41]), a 24-item measure focusing on sense of presence (e.g., “How much did your experiences in the virtual environment seem consistent with your real world experiences?”) and immersion (e.g., “How involved were you in the virtual environment experience?”). The PQ assesses global immersion and has four subscales: involvement, sensory fidelity, adaption/immersion, and interface quality. Possible PQ scores range from 0 to 175, with higher scores indicating higher presence/immersion; and the client satisfaction questionnaire-revised (CSQ-R; [42]), an 8-item measure to assess satisfaction with a product or service. This scale was modified to be study-specific to assess for the acceptability of VR-CET (e.g., “Have the services you received from your virtual reality therapy helped you to deal more effectively with your eating disorder?”). Scores on the CSQ-R range from 8 to 32, with higher scores indicating greater satisfaction. Both the SSQ and the PQ were assessed after each session, and the CSQ-R at post-treatment.

Among therapists, ratings of satisfaction delivering the intervention were obtained with a version of the CSQ-R that included study-specific modifications to be appropriate both for therapists and VR-CET (e.g., “If you were to provide services for an eating disorder again outside the study, would you use virtual reality?”). Scores on the CSQ-R range from 8 to 32, with higher scores indicating greater satisfaction. The therapist version of the CSQ-R was assessed at post-treatment.

#### 2.4.4. Preliminary Signals of Effectiveness

Preliminary signals of effectiveness were determined by changes in the frequency of disordered eating (e.g., OBEs, SBEs, and purging) from pre- to post-treatment and to follow-up and abstinence rates. Frequency of disordered eating behaviors (e.g., binges and purges) was assessed over the prior week (e.g., seven days). Binge episodes (defined as eating accompanied by a sense of loss of control) were distinguished as objective (i.e., ingesting what others would agree was an unusually large amount of food given the circumstances; OBEs) or subjective (i.e., the perception that one ate too much food at a given time but did not objectively eat a large amount according to general standards; SBEs), as defined by the eating disorder examination questionnaire (EDE-Q, [43]). After each session, study therapists were sent a study-specific post-session questionnaire to report the participant’s binge/purge frequency over the seven days prior. These clinician-derived frequencies were used for both baseline and post-treatment binge and purge data. For follow-up frequencies, a modified version of the EDE-Q [43] assessing behaviors over the prior 28 days was sent to all participants at least one month post-treatment (M = 7.10 months, SD = 2.10). Average weekly frequencies were obtained by dividing the 28 day binge and purge reports by four.

Both OBE and SBE binge abstinence and purge abstinence at post-treatment was defined as zero episodes over the prior seven days given the short duration of treatment. At follow-up, abstinence was defined as an absence of behaviors over a 28-day period.

Participants were charged for sessions at the standard clinic rate for psychotherapy. Participants received no study-related compensation for their time completing questionnaires, etc.

### 2.5. Statistical Methods

Descriptive statistics (e.g., mean, standard deviation) are presented given the small sample size of the study. Exploratory analyses using paired *t*-tests were performed between pre-, post-, and follow-up for binge-eating and purging behaviors. Both an intent-to-treat analysis (*n* = 11) and completer analysis (*n* = 9) were performed with available data [44]. We adopted a *p*-value of <0.05. Analyses were calculated using SPSS 25 for Mac. Given the exploratory nature of the study, corrections were not made for multiple comparisons.

## 3. Results

### 3.1. Participant Population

Initial participants (*n* = 11) were, on average, 40.90 (SD = 5.70) years of age, BMI = 31.80 (SD = 8.10) kg/m^2^, female (90.90%), Caucasian (72.72%), at least college educated (100%), diagnosed with BED (72.72%), reported a history of comorbid major depression (81.82%), and engaged in concurrent treatment (63.63%), such as medication management, individual, and/or group therapy for comorbid diagnoses. Please refer to Table 1 for details.

### 3.2. Feasibility

#### 3.2.1. Participants

Participant recruitment, intervention completion rates, session attendance, and questionnaire completion: Sixteen patients were referred by clinic providers to the study. All patients met eligibility criteria, as confirmed by the research coordinator. Of the 16 eligible participants, 11 (68.80%) chose to enroll. Two (18.20%) of the 11 enrolled patients dropped out (illness =1, technical difficulties VR equipment =1). Both had completed at least four sessions. Hence, nine (81.81%) completed the intervention. Of these completers, the average number of sessions attended was six out of 7–8 (SD =1.57). In terms of questionnaire completion rates, all eleven participants (100%) completed the demographic questionnaire. Completion rates for the post-session questionnaires were 62.12% (41/66) for the SSQ and 76.40% (42/55) for the PQ. Note that both the SSQ and PQ had suboptimal completion rates by participants. Participants self-reported that they did not feel they “needed to continue completing it after each intervention” if their experience did not change, therefore data were missing from both of those measures. Completion rates for the participant version of the post-treatment modified CSQ-R were 90.90% (10/11). For the follow-up questionnaire, administered at least one-month after post-treatment (mean of 7.10 (SD = 2.10) months), completion rates were 81.81% (9/11).

#### 3.2.2. Therapists

Study therapist recruitment and questionnaire completion: eight therapists attended an initial information session about VR and eating disorders. Of those eight, six (75%) chose to attend the two 2-h training workshops to become study therapists. Of these six, five (83%) actually enrolled as study therapists. All therapists were eating disorder specialists at the doctoral level (four PhD/PsyD, one MD). All were female. Therapists, using Qualtrics, completed a total of 74 post-session study notes out of possible total of 88 (74/88, 84.10%) in addition to their required medical record therapy notes. All five therapists (5/5, 100%) completed the therapist version of the post-treatment modified CSQ-R.

Overall, these data suggest the treatment and data collection were feasible to conduct for both study participants and therapists.

### 3.3. Acceptability

#### 3.3.1. Participants

Simulator sickness questionnaire: Mean total ratings for the SSQ were 21.10 (SD = 21.60), which is in range of other VR studies whose total mean SSQ scores range from 14.30–35.30 [45]. Mean nausea subscale scores were 32.10 (SD = 33.10), oculomotor subscale scores were 24.8 (SD = 24.23) and disorientation subscale scores were 15.20 (SD = 22.80). The total score range of SSQ scores for study participants were 0 (meaning no symptoms reported) to 67.30 (moderate symptoms reported).

Presence questionnaire: Participants’ mean total PQ rating was 106.40 (SD = 13.60) out of 175, indicating the VR experience was sufficiently compelling. Sensory fidelity (i.e., consistency with actual real-world experiences) also was rated highly, with a mean of 25.10 (SD = 2.80) out of 28. Mean involvement was 60.0 (SD = 6.00) out of 84 indicating a moderate sense of involvement. Both the adaptation/immersion subscale, with a mean of 21.3 (SD = 8.40) out of 42, and the interface quality subscale, with a mean of 10.6 (SD = 2.10) out of 21 received the lowest ratings. These subscale scores indicate participants did not necessarily feel they were optimally adapting to the VR exposures nor did they feel fully proficient interfacing with the VR equipment.

CSQ-R-Patients: The mean CSQ-R score was 28.00 (SD = 3.70) out of a possible 32, indicating high treatment satisfaction. Two-thirds of participants rated the quality of the VR-CET as “excellent”, with the remaining third rating it as “good”.

#### 3.3.2. Therapists

CSQ-R-therapists: Therapists’ mean CSQ score was 27.90 (SD = 4.63) out of a possible 32, indicating overall high satisfaction with delivering the intervention. Seventy-five percent reported the intervention to be either “effective” or “very effective” in helping treat their patients’ eating disorders. However, 50% reported difficulty teaching their patients how to use the VR equipment and noted feeling unsuccessful using it themselves.

Thus, the intervention was viewed as generally acceptable by both participants and study therapists based on the SSQ, PQ, and modified CSQ. However, areas of lower acceptability were noted by participants and study therapists, particularly with regard to their sense of proficiency using the VR equipment, see Table 2.

### 3.4. Preliminary Signals of Effectiveness

#### 3.4.1. Intent-to-Treat (*n* = 11)

At baseline, average OBEs/week were 3.27 (SD = 1.56). At post-treatment, OBEs were reduced by 71.30% to 0.94 (SD = 1.34)/week (*p* < 0.001). At follow-up, reported OBEs/week increased to an average of 1.20 (SD = 1.66), representing an overall significant decrease of 63.30% from baseline (*p* = 0.01) but an increase of 27.70% from post-treatment. For SBEs, participants reported an average of 3.09 (SD = 2.30)/week at baseline. By post-treatment, these episodes decreased by 58.9% to an average of 1.27 (SD = 1.06)/week (ns). In terms of post-treatment abstinence (no binge episodes over the prior week), 45.45% (5/11) of participants were abstinent from OBEs, with 18.20% (2/11) abstinent from any loss of control (LOC) eating episodes (i.e., no OBEs or SBEs). At follow-up, participants had an overall decrease of 21.40% (ns) in LOC episodes from baseline, but an increase from post-treatment of 96.70%. To see the full table, which includes information about purging and subjective binge episodes (SBEs), please refer to Appendix A.

#### 3.4.2. Completers (*n* = 9)

Completer analyses were similar to those for intent-to-treat. Mean OBE episodes/week significantly decreased by 84.20%, or from 3.80 (SD = 1.20) at baseline to 0.60 (SD = 1.00) at post-treatment (*p* < 0.001). At follow-up (which was conducted on average 7.10 (SD = 2.10) months post-treatment) participants reported an average of 1.22 (SD = 1.80) OBEs/week, an overall decrease from baseline of 67.90% but increase from post-treatment of 103.30% (ns). SBEs followed a similar (though not significant) trend, decreasing 59.30% from a baseline mean of 3.00 (SD = 2.54) SBEs/week to a post-treatment mean of 1.22 (SD = 1.14) SBEs/week (ns). At follow-up, SBEs/week averaged 2.40 (SD = 2.56), an overall significant decrease from baseline of 20% (*p* < 0.05) but increase from post-treatment of 96.70%. In terms of abstinence, 55.60% (5/9) of completers were abstinent from OBEs at post-treatment and 22.22% (2/9) abstinent from both OBEs and SBEs. At follow-up, 22.22% (2/9) of completers were abstinent from OBEs over the prior 28 days and 22.22% (2/9) had neither OBEs nor SBEs over the prior 28 days. At baseline, only two participants were actively purging; both were treatment completers. Their baseline purge frequency was 1.00 (SD = 1.41)/week, decreasing to 0/week at post-treatment, a 100% reduction, which was maintained through follow-up. To see the full table, please refer to Appendix A.

## 4. Discussion

This small, uncontrolled pilot study provides preliminary evidence that immersive VR-CET for binge eating delivered in a U.S. clinic setting is feasible, generally acceptable, and possibly effective. Although previous research [20,21] demonstrated the efficacy and superiority of VR-CET as a second-level treatment strategy compared to additional CBT (A-CBT) for refractory patients (patients diagnosed with BN or BED who were initially treated unsuccessfully with a structured CBT treatment), these studies were conducted in highly controlled research settings. The current study is the first to: (1) translate a European research-based VR-CET protocol to a culturally adaptive, and practice ready intervention; and (2) evaluate feasibility and acceptability of VR-CET for binge eating in a real-world outpatient clinic.

### 4.1. Feasibility

Therapists: Therapist interest in learning a novel VR intervention was fairly high with 63% (5 of 8) of invited therapists joining the study. It is worth noting that therapists involved in this study volunteered unprotected time to participate in training as they viewed it as valuable to enhance their clinical skill set, however, they did receive a small monetary bonus for this time ($400/patient paid by the study grant). It is possible that therapist interest may have been higher if all study activities (e.g., learning how to use VR equipment) were time protected (as reflected in productivity reports) and financially compensated. Even without protected time, completion rate of study notes was 84.10%, with 100% completing the post-treatment modified CSQ-R.

Participants: Nine (82%) participants completed treatment and two dropped (18%) after an average of four sessions. This drop-out rate was higher than the previous VR-CET study showing no participant dropout, though that study was conducted in a research setting with no costs to participants [20]. Of note, one of the two dropouts was due to an injury requiring suspension of treatment and the other was related to dissatisfaction regarding technical issues with the VR equipment. This patient’s therapist also had difficulty with the Oculus Rift technology due to competing demands and a lack of protected time to address technological issues rendering receipt of IT support challenging. As such, the patient’s experience likely reflected their therapist’s experience. However, our presence data did indicate that participants did not feel completely proficient using the VR equipment. Unfortunately, due to a lack of research reporting overall scores on the presence questionnaire, it is difficult to compare these results to other VR interventions. As VR technologies become easier to use (and less expensive), drop out related to technical issues likely would be lower. However, an 18% drop out rate falls approximately mid-range for treatment trials for BED (4–34%) [46,47,48,49]. In general, these data suggest the adoption of VR-CET appears feasible within a real-world clinic.

Study questionnaires were completed by both participants and therapists, and most questionnaires had completion rates between 80 and 100%. Evaluating feasibility of uncompensated survey completion is important in a real-world study as this process is similar to measurement-based care (MBC) protocols being promoted [50] and implemented across clinics [51]. MBC is defined as the practice of basing clinical care on client data collected throughout treatment [52]. The benefits of systematic data collection include insight into treatment progress, highlighting ongoing treatment targets, reduced symptom deterioration, and improved client outcomes [52,53]. More specifically, adding MBC to usual care can result in significant improvement in treatment outcomes and active involvement of clients in the treatment process [52]. For clinicians, MBC can provide important information about targets for clinician intervention and enhance the accuracy of clinician judgments by providing an objective assessment of client treatment progress. Thus, the high completion rates for study questionnaires suggest that the number of surveys administered was generally acceptable to participants and tracking VR-CET outcomes in clinic settings is feasible.

### 4.2. Acceptability

Participants: Although many participants experienced some symptoms of simulator sickness, the global SSQ average score was similar to other studies and lower than those using VR gaming content or 360° videos [45]. Furthermore, other than slight to moderate general discomfort, experiences of simulator sickness did not lead to notable adverse effects or treatment dropout. Participants rated the intervention as compelling, in that it was viewed as both immersive and involving (i.e., the VR environments felt consistent with actual real-world experiences). Mean CSQ-R scores were high at 28.00 out of a possible total score of 32.

Therapists: Therapist CSQ-R scores were high at 27.90 out of a possible total score of 32. Our results indicate that VR-CET also appears to be acceptable within a clinic setting.

### 4.3. Effectiveness

Our preliminary signal of effectiveness is noteworthy in a sample of patients who had not completely responded to previous eating disorder treatment. All participants showed reductions in binge eating within just about seven sessions, with a post-treatment objective binge abstinence rate of 55% for completers, which is similar to that reported in previous VR-CET studies (53%, *N* = 32; [20]) and BED trials with larger samples (50.9% total weighted percentage; [11]). Of note, definitions of post-treatment abstinence rate do vary across these trials. Among those purging at baseline, purge episode frequency decreased from one to zero episodes per week, with 100% abstinence maintained at follow-up. Previous VR-CET for binge eating studies reported that 75% of patients with BN (*N* = 16) achieved abstinence from purging episodes post-treatment [20], with 73.3% abstinence maintained at 6-month follow-up [21].

Overall, the observed reductions in eating disorder symptomatology at post-treatment and follow-up indicate preliminary evidence of effectiveness of VR-CET in a real-world clinic. However, the durability of treatment effects for binge eating may be a concern. Future iterations of the protocol may wish to consider a means for improving the maintenance of effects, such as with booster sessions or a self-administered session at home.

Future work to understand the mechanisms of action is required. Based on recent approaches to exposure therapy, one hypothesized mechanism of action to be tested is the inhibitory learning model of extinction [27,34]. The inhibitory learning model maximizes extinction learning by designing exposures to highlight expectancy violations and consolidate the non-occurrence of the anticipated feared event [27]. It is possible that VR-CET is an effective way to enhance inhibitory learning as the VR simulates immersive real-world experiences, allowing the therapist and patient to focus on the urge to binge, while preventing binge eating as the foods are not physically available to consume.

### 4.4. Cultural Adaptations

A unique contribution of this study is the development of the first practice-ready VR-CET therapist manual, one that can be adapted for ongoing clinical use and research across clinics and cultural environments. The adaptations to and observations from translating a research-based VR-CET program from Europe into a real-world clinical setting in the U.S. are worth emphasizing. We adapted the original European food and environment list to be culturally relevant for a U.S. population and, although changes could be considered minor, adaptations were important to the success in eliciting cravings and urges in our patients. Clinics in other countries similarly may need to adapt protocols to ensure cultural relevance. Additionally, although the original research protocol required twice weekly therapy sessions, therapists in our real-world clinic setting appreciated the option of twice or once weekly therapy, based on the patient’s clinical presentation (e.g., severity of symptoms) or logistical factors (e.g., whether the patient could attend twice weekly in-person appointments), as typical of outpatient clinical settings. Given the change in session frequency appeared important for feasibility coupled with the fact that it did not appear to drastically alter treatment effects, offering clinic therapists the option to set/alter session frequency from twice to once per week, as clinically indicated, appears important. Our addition of out-of-session exposure work as part of our homework may have compensated for reduced intensity (e.g., from 2 to 1/week) of the in-session exposures, as the prior European protocol did not include homework. Despite the once-weekly frequency, the treatment remains time efficient to deliver, with an average of seven sessions. Future studies may test the optimal dose of treatment.

Another adaptation was the therapists’ use of strategies to increase the anxiety response and maximize the participants’ immersion in the virtual environment (e.g., emotional priming). Therapists had observed that merely viewing the foods in the virtual environment did not always produce high anxiety or cravings (e.g., some participants found the VR food unrealistic, not relevant enough to their specific binge foods, or reported that they did not experience urges unless experiencing emotional distress). In these situations, therapists used participants’ history and/or emotional priming techniques to enhance anxiety about losing control. For example, therapists elicited participants’ memories related to the foods, invited participants to manipulate the food in the environment in various ways, and asked participants to focus on sensory experiences, thoughts, and/or emotional reactions to the food. We hypothesized that these techniques may be especially helpful for those who endorse emotional eating (eating in response to intense emotions rather than visual cues). The relationship between eating behavior style (emotional, restrictive, and external) and cravings and anxiety to food-related VR environments in patients with BN and BED has been previously studied [31]. External eating has been shown to predict cue-elicited craving, whereas emotional and external eating have been shown to predict cue-elicited anxiety [31]. As such, both craving and anxiety ratings are used to create the 13-step food–environment hierarchy in the existing VR-CET for binge eating protocols. However, future research should test whether VR-CET’s effects vary by eating behavior style. The Dutch eating behavior questionnaire (DEBQ; [54]), used for the assessment of restrictive, emotional, and external eating behavior, may be a useful measure to consider in future research to examine eating styles. Further, future research may wish to test whether VR-based emotion regulation interventions are of specific utility among patients with an emotional eating style. Our team has developed and is currently testing a brief VR-based emotion regulation intervention for individuals with emotional eating; due to COVID-19, the 3-D intervention was delivered remotely, via 2-D screen sharing.

### 4.5. Limitations

The present study has several limitations. First, the sample size was small; thus, only descriptive statistics and exploratory analyses of significance were used. As such, the results should be interpreted with care. Second, there was a lack of diversity in terms of demographics. Third, study participants were allowed to continue concurrent treatment for concerns other than binge eating, consistent with the flexibility available to patients in real-world settings. As such, not all observed changes may be attributable to VR-CET. Fourth, therapists were able to rely on their clinical judgment and use of basic therapeutic strategies while delivering the intervention, including sound individualized case conceptualization, Socratic questioning and teaching coping skills to patients with eating disorders. Therapists were expected to personalize therapeutic content to their individual participants. Therapists also provided two kinds of homework: (1) additional in-vivo, out-of-session exposure-based assignments at home or in other settings (required); and (2) practicing coping skills (optional) to help the patient strengthen and generalize application. Although inviting therapists to use their own unique style is consistent with real-world outpatient care, it should be noted that each participant likely received a slightly different experience and that therapists (all doctoral level) drew on their previous training in eating disorders and CBT to implement the manual skillfully. The degree of training needed to implement this manual effectively is unknown at this time. Fifth, this “real-world” study received grant funding to pay for the VR computers, associated equipment, and software development. In other university affiliated clinical settings, departments might be willing to provide funding for clinician training and the purchase of needed technologies. Finally, this study was not designed to accurately assess total costs and time to train providers. These metrics are important to measure in future work in order to accurately assess feasibility and cost-effectiveness in a real-world setting. In addition, there are pros and cons to both immersive and semi-immersive programs and future research should continue to investigate if there are significant differences in effectiveness [16].

In conclusion, despite the aforementioned limitations, we believe this uncontrolled pilot study provides a valuable contribution to the scant literature investigating the feasibility and acceptability of translating innovative studies described in the research literature into practice-ready interventions in real-world clinic settings. The findings from this uncontrolled pilot study provide preliminary evidence that immersive VR-CET for binge eating among patients who remain symptomatic after prior treatment is generally feasible, acceptable, and shows a promising signal of effectiveness within a U.S. clinic setting.

## Figures and Tables

**Figure 1 jcm-10-01511-f001:**
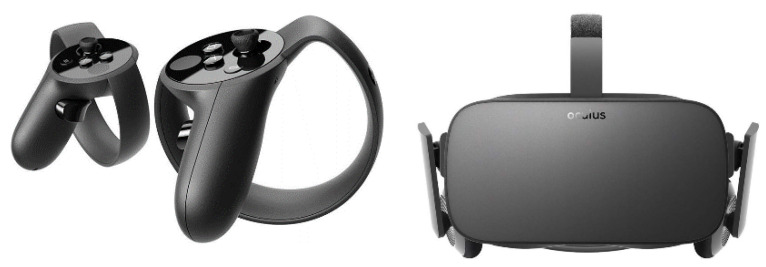
Head mounted display and controllers.

**Figure 2 jcm-10-01511-f002:**
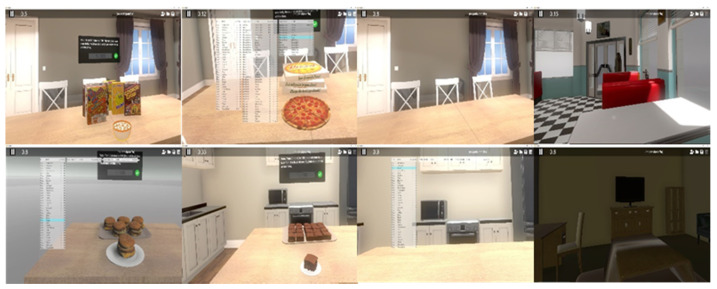
Virtual environments and foods.

**Table 1 jcm-10-01511-t001:** Participant characteristics.

**Characteristics**	***n* = 11 (100%)**
Age, *M (SD)*	40.90 (15.7)
BMI, *M (SD)*	31.80 (8.10) kg/m^2^
Range	21.80–43.60 kg/m^2^
**Gender Identity, 11 (100%)**	
Male	1 (9.10%)
Female	10 (90.90%)
**Race, 11 (100%)**	
Black or African American	1 (9.10%)
Caucasian	8 (72.72%)
Asian	2 (18.28%)
**Education, 11 (100%)**	
Bachelor’s Degree	6 (54.50%)
Some Graduate School	1 (9.10%)
Completed Graduate School	4 (36.40%)
**Eating Disorder Diagnosis, 11 (100%)**	
Binge-Eating Disorder (BED)	8 (72.70%)
Bulimia Nervosa (BN)	3 (27.30%)
**Comorbid Diagnoses, 11 (100%)**	
Hx Major Depression	9 (81.82%)
Hx Bipolar II	1 (9.10%)
Hx Obsessive Compulsive Disorder	1 (9.10%)
**Concurrent Treatment, 11 (100%)**	
(e.g., medication management, individual, and/or group therapy)	
Yes	7 (63.63%)
No	4 (36.36%)

**Table 2 jcm-10-01511-t002:** Acceptability metrics.

**SSQ (Range = 0–235.6)**	**Mean, Standard Deviation**
Range of scores	0–67.30
Total (*M, SD*)	21.10 ± 21.60
Nausea (*M, SD*)	32.10 ± 33.10
Oculomotor (*M, SD*)	24.80 ± 24.20
Disorientation (*M, SD*)	15.20 ± 22.80
**PQ (Range = 0–175)**	
Range of scores	94–134
Total (*M, SD*)	106.40 ± 13.60
Involvement (*M, SD*) Range: 0–84	60.00 ± 6.00
Sensory Fidelity (*M, SD*) Range: 0–28	25.10 ± 2.80
Adaptation/Immersion (*M, SD*) Range: 0–42	21.30 ± 8.40
Interface Quality (*M, SD*) Range: 0–21	10.60 ± 2.10
**CSQ-Patient (Range = 8–32)**	
Range of scores	20–31
Total (*M, SD*)	28.00 ± 3.70
How would you rate the quality of your virtual reality therapy?	Excellent (66.70%) Good (33.30%)
Would you recommend this treatment to a friend that was struggling with an eating disorder?	Yes, Definitely (66.70%) Yes, I Think So (33.30%)
**CSQ-Therapist (Range = 8–32)**	
Range of scores	18–31
Total (*M*, *SD*)	27.90 (4.60)
How easy was it to teach your patient to use the virtual reality system?	Easy (50%) Difficult (50%)
How successful do you feel at using the technology of the virtual reality system?	Not at all successful (50%) Successful (50%)
How effective do you believe the virtual reality therapy was in helping treat your patient’s eating disorder?	Very Effective (25%) Effective (50%) No difference (25%)

## Data Availability

The data presented in this study are available on request from the corresponding authors. The data are not publicly available due to the small sample size and confidentiality concerns.

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
