# Peer review of "Translating Virtual Reality Cue Exposure Therapy for Binge Eating into a Real-World Setting: An Uncontrolled Pilot Study"

_jcm, 2021, doi:10.3390/jcm10071511_

Round 1

Reviewer 1 Report

Thank you for the opportunity to review this manuscript. I think it is a very interesting and current topic. The application of virtual reality-based technology is a growing field and it is important to explore its usefulness for people with eating disorders.
Below are some changes that may enhance the comprehensibility of the manuscript:

INTRODUCTION
The introduction does not explain the definition of Virtual Reality applied to the clinic. It would be useful to specify what the technique used consists of. That is, what is virtual reality and the types of virtual reality that are used for the treatment of people with eating disorders. I mean immersive, semi-immersive virtual reality.

METHODOLOGY (2.3.2 Assessment and intervention)
Add an image of the device used and/or the screens. Some of the images that are part of the supplementary material could be useful to introduce in the text.

Reviewer 2 Report

Thank you very much for the opportunity to review the article entitled "Translating Virtual Reality Cue Exposure Therapy for Binge Eating into a Real-World Setting: A Pilot Study." The article is characterized as a brief report (or short communication), therefore it will be reviewed as such. The application of virtual reality in medicine is booming and many more years of research lie ahead. Due to showing some effectiveness in psychiatry, research of this type is extremely important. Therefore, I am glad that the authors have taken up this topic. The main aims of this study are to determine the feasibility of VR-CET in clinical practice and to determine the effectiveness of this therapy under certain conditions. However, this article has several inaccuracies. First of all, it is worth specifying the design of this study. So far, this is a discrepancy between the pilot study and the case series. Secondly, the discussion needs to be supplemented and reorganized. In my opinion, it will be more convenient if the authors divide it into sections. Third, the lack of a control group makes it impossible to fully objectively determine the effectiveness. On the other hand, it also provides very interesting information. Therapists' attitude towards therapy is certainly a value. The assessment of side effects is also a strength of this article. In my opinion, this publication provides a lot of new knowledge on the subject. However, I would like the authors to comment on my questions/suggestions below.

Comments:

  1. The idea behind head-mounted display VR is an immersion experience by the participant. Did the assessment of anxiety symptoms 13 times over 45 seconds not make the participant feel immersed?
  2. Does this study meet all the assumptions necessary for the study to be called a pilot study? From the discussion, the authors begin to call it a case series study. In this case, does the study meet the assumptions of the case series design? Or maybe it's a mix of the two? Please specify the study design.
  3. In the title of the study, BED is defined as the target group, however, 3 people with BN were also included. Wouldn't it be advisable to either generalize the title or add a BN to it?
  4. Correct me if I'm wrong, but from an hour of therapy called VR-CET, the 30 min maximum included VR, and the rest was devoted to other elements of the therapy (line 154). So it was a mixed intervention with elements of CBT and VR? The purpose of this type of study is to determine whether such interventions can be carried out under more controlled conditions in the future.
  5. Line 60 - I do not think it is necessary to indicate that the recruitment took place before COVID-19.
  6. I propose to divide the discussion into two parts: concerning the effectiveness of therapy and the opinions of therapists. At this point, the two parts are mixed up, which makes interpretation difficult. Perhaps you might want to do that in the results as well.
  7. What more information does this study provide compared to the previous study of this device (apart from the setting of the study outside UE): “A Randomized Controlled Comparison of Second-Level Treatment Approaches for Treatment ‐ Resistant Adults with Bulimia Nervosa and Binge Eating Disorder: Assessing the Benefits of Virtual Reality Cue Exposure Therapy ”?
  8. In the discussion, I miss delving into the likely mechanisms of the results obtained. Why was the applied therapy so effective in this group?
  9. Correct p-value reporting. One time you report with spaces, one time without, another time with zero (line 336), another time without (line 324). Make it uniform.
  10. Line 365-376 - is describing the MCB relevant in the context of this study?
  11. Line 398 - correct 360o.
  12. Unify the number of decimal places.

Reviewer 3 Report

Even for a pilot study, this study itself and the manuscript lack appropriate scientific rigour.

The overall rationale is not clear, the authors do not present a case for why the gaps in treatment should be addressed with VR-CET. The fact that one can use such technology does not in itself justify its use unless there are appropriate clinical or research-based reasons for doing so. This is not established in the introduction. Why is the exposure in VR necessary? Why can it not be done in real-life? Why CGI stimuli rather than real stimuli? All of these things need to be covered for a convincing introduction.

The manuscript is not accompanied by any images of the stimuli. This is a huge problem in VR literature, where frequently the stimuli are of poor quality and the authors simply do not show it. In fact, in lines 434 onwards it is specifically addressed that the therapists explained the that stimuli was not compelling enough. The authors should focus their future work on improving the realism and look of their technique first, before trying to push for application.

The results overall are meaningless, in the absence of a control group it is impossible to know whether any differences in data are due to the exposure or the regular treatment the participants were receiving alongside the VR exposure. It is entirely possible that the observed decreases in binging episodes would be observed after weeks of regular treatment regardless of this VR exposure. It is imperative that the authors run appropriately controlled studies.

It is an over-statement that this technique is ‘feasible, acceptable and possibly efficacious’ (line 351). The fact that it is feasible is meaningless, anything can be feasible, the important question is whether it works. It also does not seem acceptable, both the patients and the therapists reported difficulties, to the point of one participant dropping out because of it. The participants did not find it compelling when reporting on the questionnaires. The fact that they said they would recommend this to friends is meaningless, as it may be an artefact of doing something ‘more’ or ‘extra’ than just talking therapy.

Overall, the tone of the paper is not rigorously scientific; it is written as a sales pitch. It is full of over-statements, e.g. line 358/9 or 348. If something is worth doing, it is worth doing well. The authors should polish this technique (e.g. the look of stimuli, appropriate training for the therapists), then run a properly designed scientific study with sufficient sample size and a control group. Then test this technique.

Some more specific comments:
Line 43. Please provide a more robust argument for why novel treatments are needed to improve outcomes and sustain recovery. Why is the improvement of extant treatment techniques an option?

Lines 52-56: break up overly-long, run-on sentence

Line 59: The sample size is incredibly small, while I appreciate that this is a preliminary study, it is difficult to run any statistical analyses in a sample this small.

Line 62: what is the ‘standard diagnostic evaluation?’, specify

Section 2.3.1: if two session per week are ‘basic’ format, the choice of naming the version with one session per week as ‘extended’ is unclear.

Line 115: are any objective physiological responses measured? E.g. heart rate, galvanic skin response?

Lines 116-118: “Through repeated exposure to stimuli and response prevention, patients weaken the response and meaning to cues by disconfirming their expectancies and increasing their tolerance of aversive emotions.” This is rather vague and non-descript, amend to be informative.

Why is the publication not accompanied by any images of the stimuli? This is a huge problem in VR literature, where frequently the stimuli are of poor quality. Especially as authors are describing how changes have been made to existing environments – show the reader the changes.

Lines 150-151: elaborate

Line 157: imaging what food would taste like seems inadequate to ‘engage other sensory modalities’; this seems like an overstatement.

Line 158-159: how do the therapists check for under-engagement or avoidance?

Lines 298-305: It is overall discouraging that the participants did not report feeling immersed in the exposure experience. The heart of VR-based work is its immersive and interactive nature. It seems that this technique has failed to deliver on this; yet again questioning the validity of this work. Is it that the stimuli and environments are not compelling? It is hard to say, as pointed out before, as the authors did not provide any images to show their intervention.

Reviewer 4 Report

This paper provides first evidence of the feasibility and possible efficacy of a VR-based cue-exposure therapy on a small sample of previously treated BED and BN patients. 

I found this article really interesting and well-written, I enjoyed reading it! It contributes to bridge the gap about the scientific literature about the effectiveness of VR-CET among EDs patients who are especially resistant to treatment. Furthermore, the authors propose a technology that can be easily applied in the clinical setting.

I have just few minor suggestions:

Introduction:

I would propose to the authors to insert a brief paragraph with the aim to briefly presenting/describing VR-CET.

Results:

  • Line 328: I feel like if results about SBEs at follow-up are missing.
  • I would add in the text a reference to Table S2.
  • Regarding Table S2, I would suggest specifying which table is referring to completers and which one to intent-to-treat.

Round 2

Reviewer 3 Report

The manuscript is much improved. I would implore the authors to gather larger sample sizes in future work to improve the rigour of the study. Work on clinical application needs very rigorous and careful scientific consideration.